# Reinforcement learning on structure-conditioned categorical diffusion for protein inverse folding

## Abstract

Protein inverse folding—that is, predicting an amino acid sequence that will fold into the desired 3D structure—is an important problem for structure-based protein design. Machine learning based methods for inverse folding typically use recovery of the original sequence as the optimization objective. However, inverse folding is a one-to-many problem where several sequences can fold to the same structure. Moreover, for many practical applications, it is often desirable to have multiple, diverse sequences that fold into the target structure since it allows for more candidate sequences for downstream optimizations. Here, we demonstrate that although recent inverse folding methods show increased sequence recovery, their "foldable diversity"—i.e. their ability to generate multiple non-similar sequences that fold into the structures consistent with the target—does not increase. To address this, we present RL-DIF, a categorical diffusion model for inverse folding that is pre-trained on sequence recovery and tuned via reinforcement learning on structural consistency. We find that RL-DIF achieves comparable sequence recovery and structural consistency to benchmark models but shows greater foldable diversity: experiments show RL-DIF can achieve an foldable diversity of 29% on CATH 4.2, compared to 23% from models trained on the same dataset. The PyTorch model weights and sampling code are available on GitHub.

## 1 Introduction

The task of designing sequences of amino acids that fold into a desired protein structure, also known as protein "inverse folding" (IF) Yue & Dill (1992); Ingraham et al. (2019), is a critical step for many protein design applications in areas such as therapeutics, biomaterials, and synthetic biology Ferruz et al. (2023); Cao et al. (2022); Dickopf et al. (2020). Deep learning-based IF models are typically trained to recover the observed sequence $S$ of a protein, when conditioned on the corresponding backbone structure $X$. Models are then evaluated on their ability to recover the original sequence ("sequence recovery"), consistency of the generated sequence's (usually predicted) structure with the target structure ("structural consistency"), and the diversity of generated sequences ("sequence diversity"). Sequence diversity is of particular interest since the inverse folding problem is a one-to-many mapping in which a single (input) structure could be formed by multiple (output) amino acid sequences. For practical applications, it is often desirable to find these alternative sequences that can fold into the desired structure, since it enables greater breadth of options for critical downstream optimization steps such as improving stability, preventing aggregation, and reducing immunogenicity. However, methods of increasing sequence diversity (e.g. raising the sampling temperature of an autoregressive model) may degrade the structural consistency of candidates. Hence, a useful property of a protein inverse folding model is its "foldable diversity"—i.e. the diversity of generated sequences that fold into the target structure.

In this paper, we first characterize the trade-off between sequence diversity and structural consistency across multiple IF methods. We use these observations to motivate foldable diversity as a new metric to measure performance of IF models.

We then report a new protein inverse folding model, RL-DIF (Figure S3). RL-DIF is a categorical diffusion model pre-trained on sequence recovery and fine-tuned with reinforcement learning (RL)

to optimize structural consistency. The categorical diffusion component of our method is inspired by GradeIF Yi et al. (2023a) and PiFold Gao et al. (2022), and is trained on observed protein structures and sequences from the CATH 4.2 dataset. We then fine-tune the diffusion model with denoising diffusion policy optimization (DDPO) Black et al. (2023) to maximize the expected structural consistency of designed sequences. We find that this two-phase training results in higher-quality designs while simultaneously ameliorating the tension between sequence diversity and structural consistency.

We evaluate state of the art IF models and RL-DIF on four benchmark datasets: CATH 4.2, TS50, TS500, and CASP15, and show that our model is able to balance sequence diversity and structural consistency resulting in an increase of foldable diversity (29% with RL-DIF compared to 23% with the best prior model), whilst keeping structural consistency at competitive levels to SOTA inverse folding methods.

## 2 PRELIMINARIES

### 2.1 DISCRETE DENOISING DIFFUSION PROBABILISTIC MODELS

The task of protein inverse folding is to design an amino acid sequence $S$ compatible with spatial coordinates $X \in \mathbb{R}^{N \times 3}$ representing the positions of the $C\alpha$ atoms in the backbone of a protein with $N$ residues. We represent $S$ as a sequence of one-hot vectors with vocabulary $\mathcal{V}$ consisting of the 20 naturally-occurring amino acid species. That is, $S \in \{0, 1\}^{N \times |\mathcal{V}|}$ subject to $\sum_j S[i, j] = 1, \forall i$.

As introduced in GradeIF Yi et al. (2023b), we use conditional discrete denoising diffusion probabilistic models (D3PM) Austin et al. (2023) to model $p(S|X)$. Formally, we define a forward Markov diffusion process as random variables $S_0, \ldots, S_T$ related by:

$$S_t \sim q(S_t|S_{t-1}, S_0) \equiv \text{Cat}(S_t; p = S_{t-1}Q_t) \tag{1}$$

where $Q_1, \ldots, Q_T$ is a sequence of $|\mathcal{V}| \times |\mathcal{V}|$ transition matrices, and $S_0 \equiv S$ is the sequence observed in nature. Defining $\bar{Q}_t = \prod_{k=1}^t Q_k$, the posterior is:

$$q(S_{t-1}|S_t, S_0) = \text{Cat}\left(S_{t-1}; p = \frac{S_t Q_t^\top \odot S_0 \bar{Q}_{t-1}}{S_0 \bar{Q}_t S_t^\top}\right) \tag{2}$$

It is standard to choose $Q$s such that $q(S_T|S_0)$ is a stationary distribution $p(S_T)$. In this case, learning a reverse Markov process $p_\theta(S_{t-1}|S_t; X)$ allows us to generate novel sequences by first sampling $S_T \sim p(S_T)$ and then iteratively denoising samples with $p_\theta$.

### 2.2 DENOISING DIFFUSION POLICY OPTIMIZATION

Assume we can assign a scalar reward $R(\hat{S})$ for every sample $\hat{S}$ from a diffusion model $p_\theta$. DDPO Black et al. (2024) treats the reverse denoising process as a $T$-step Markov decision process and defines a policy gradient on $p_\theta$ to maximize the expected reward $\mathcal{J}(\theta)$:

$$\mathcal{J}(\theta) = \mathbb{E}_{X \sim p(X), \hat{S} \sim p_\theta(\hat{S}|X)}\left[R(\hat{S})\right] \tag{3}$$

where $p_\theta(S_0|X) = \sum_{S_1, \ldots, S_T} p(S_T) \prod_{t=1}^T p_\theta(S_{t-1}|S_t, X)$. The corresponding policy gradient $\nabla_\theta \mathcal{J}(\theta)$ is:

$$\nabla_\theta \mathcal{J}(\theta) = \mathbb{E}_{X \sim p(X), \hat{S}_0, \ldots, \hat{S}_T \sim p_{\text{old}}}\left[\sum_{t=1}^T \frac{p_\theta(S_{t-1}|S_t, X)}{p_{\text{old}}(S_{t-1}|S_t, X)} \nabla_\theta \log p_\theta(S_{t-1}|S_t, X) R(S_0, X)\right] \tag{4}$$

where the importance sampling ratio of the target and sampling policies $\frac{p_\theta(\cdot)}{p_{\text{old}}(\cdot)}$ allows for multiple optimization epochs per sample.

### 2.3 INVERSE FOLDING METRICS

To assess the quality of designed sequences, a number of metrics are used in the protein design literature. Here, we focus our discussion on three commonly-used ones. In these definitions, we

assume we have a protein structure $X$ with $N$ residues, an observed sequence $S$, and designed sequences $\{\hat{S}^1, \cdots, \hat{S}^M\}$.

- **Sequence recovery**: The average number of amino acid positions in agreement between the observed and proposed sequences:

$$\text{RECOVERY}(\hat{S}^j; S) = \frac{1}{N} \sum_{i=1}^{N} \mathbb{1}\left[ S[i] = \hat{S}^j[i] \right].\tag{5}$$

- **Self-consistency TM score (sc-TM)**: The template modeling (TM) score Zhang & Skolnick (2004) measures the similarity of two protein structures. The score is 1 if they are identical and tends to 0 for dissimilar structures. sc-TM is defined as Trippe et al. (2022):

$$\text{sc-TM}(\hat{S}^j; S) = \text{TM-SCORE}(\text{FOLD}(\hat{S}^j), \text{FOLD}(S))\tag{6}$$

where FOLD is any protein folding algorithm such as AlphaFold2 Jumper et al. (2021) or ESMFold Lin et al. (2023). Note that we compare designs to $\text{FOLD}(S)$ (not $X$) to reduce the effect of biases of the folding algorithm Gao et al. (2023).

- **Sequence diversity**: The average fraction of amino acids that differ between pairs of designs:

$$\text{DIVERSITY}(\{\hat{S}^1, \ldots, \hat{S}^M\}) = \frac{2}{NM(M-1)} \sum_{j=1}^{M} \sum_{k=1}^{j-1} \sum_{i=1}^{N} \mathbb{1}\left[ \hat{S}^j[i] \neq \hat{S}^k[i] \right]$$

$$= \frac{2}{M(M-1)} \sum_{j=1}^{M} \sum_{k=1}^{j-1} d_{\text{H}}(\hat{S}^j, \hat{S}^k).\tag{7}$$

where $d_{\text{H}}$ is the Hamming distance. We note that sequence diversity alone is not a sufficient measure of a IF method's quality, as it can be increased arbitrarily at the expense of sample quality (e.g. as measured by structural consistency).

## 3 METHODS

### 3.1 FOLDABLE DIVERSITY

In realistic protein design scenarios, it is necessary to have a diverse pool of candidates for synthesis and experimental characterization. We are particularly interested in the sequence diversity of IF models when restricted to faithful structures, and so we propose **foldable diversity (FD)**:

$$\text{FOLDABLE DIVERSITY}(\{\hat{S}^1, \ldots, \hat{S}^M\}) =$$

$$\frac{2}{M(M-1)} \sum_{j=1}^{M} \sum_{k=1}^{j-1} \left( d_{\text{H}}(\hat{S}^j, \hat{S}^k) \times \mathbb{1}\left[ \min\left\{ \text{sc-TM}(\hat{S}^j; S), \text{sc-TM}(\hat{S}^k; S) \right\} > \text{TM}_{\min} \right] \right)\tag{8}$$

Foldable diversity only considers the diversity between pairs of sequences that are both structurally consistent with the input protein backbone. Unlike sequence diversity, it is not possible to increase FD by sampling high-entropy sequences, without also ensuring they fold correctly. We therefore argue that FD is a more grounded metric for evaluating IF methods than sequence recovery (which penalizes high-quality, highly-diverse sequences) and sequence diversity (which does not consider sequence quality).

Unless otherwise stated, we set $\text{TM}_{\min} = 0.7$. Among observed protein structures, this threshold typically guarantees two proteins share a topological classification Xu & Zhang (2010); Wu et al. (2020), i.e. are structurally similar. In this regime, FD combines sequence diversity and structural consistency into a single quality measure, if we assume that all sequences with sc-TM above $\text{TM}_{\min}$ are "equally good".

## 3.2 D3PM FOR INVERSE FOLDING

As in Austin et al. (2023); Yi et al. (2023b), we train a neural network $\hat{p}_\theta(S_0|S_t)$ to recover the posterior by integrating over the vocabulary:

$$p_\theta(S_{t-1}|S_t) \propto \sum_{v \in \mathcal{V}} q(S_{t-1}|S_t, v)\hat{p}_\theta(v|S_t) \tag{9}$$

**Model Architecture**   We parameterize the network as a modified PiFold Gao et al. (2022) architecture, adding multilayer perceptrons (MLPs) to process the partially-noised amino acid sequence and diffusion timestep.

Specifically, given a set of backbone coordinates $\mathbf{X} \in \mathbb{R}^{4N \times 3}$, we first construct a kNN graph ($k = 30$) between residues. We then use the PiFold featurizer to define node and edge features $h_V$ and $h_E$. These features include distances between atoms, dihedral angles, and direction vectors.

The denoising model is a function of $h_P$, $h_E$, the partially-denoised sequence $s_t$, and timestep $t$. We use the following architecture:

$$h'_V, h'_E = \text{MLP}(h_V), \text{MLP}(h_E)$$
$$h_o = \text{MLP}([s_t, t])$$
$$h_{Vs} = [h'_V, h_o]$$
$$h_V^{out}, h_E^{out} = (10 \times \text{PiGNN})(h_{Vs}, h'_E)$$
$$p(s_0|s_t) = \text{MLP}([h_V^{out}, h_{Vs}])$$

where $[a, b]$ represents concatenation and PiGNN is the GNN layer introduced by PiFold. Unless otherwise noted, we set all hidden layer sizes to the recommended values in Gao et al. (2022).

**Training**   We observed improved performance from using the full D3PM hybrid loss, as compared to the cross-entropy loss used by Grade-IF. We use uniform transition matrices, since we did not observe a benefit from the Block Structured Matrices Henikoff & Henikoff (1992) used in Grade-IF.

The model was trained on structure-sequence pairs from the CATH 4.2 dataset with the train/validation/test split curated by Ingraham et al. Ingraham et al. (2019) based on CATH topological classifications. This dataset and split is used by most prior IF models. It comprises 18025 samples for training, 1637 for validation and 1911 for testing.

We used the Adam optimizer Kingma & Ba (2017), a learning rate of $10^{-3}$, an effective batch size of 64 (distributed over 4 Nvidia A10 GPUs), and 150 diffusion time steps. We trained for 200 epochs.

## 3.3 RL-REFINEMENT OF INVERSE FOLDING MODELS

As noted above, inverse folding is a one-to-many mapping problem, with potentially a large number of sequences satisfying a conditioned structure. Furthermore, publicly-available protein structure data represents a sparse and heterogeneous sampling of the desired distribution $p(S_0 = S|X)$. This one-to-many mapping and data paucity are key challenges in the generalization of conditional generative models Yi et al. (2023b); Dauparas et al. (2022a); Hsu et al. (2022). While collecting more data is possible, this is an expensive and slow process.

On the other hand, given a proposal sequence $\hat{S}$, we may verify its suitability for the conditioned structure $X$ by evaluating the self-consistency TM score. We therefore propose a second phase of training, in which the inverse folding diffusion model is optimized for $\mathcal{J}(\theta) = \mathbb{E}_{\hat{S} \sim p_\theta}[\text{sc-TM}(\hat{S}; S)]$. In particular, we use the DDPO algorithm summarized in Section 2.2.

**Training**   During the second phase of training, we use the same training dataset described in Section 3.2. Each training step takes as input a batch of 32 protein backbone structures. First, we sample 4 sequences per structure from the diffusion model. Raw rewards (sc-TM) are standardized ($\mu = 0, \sigma = 1$) separately for each structure: each set of 4 sequences are shifted and scaled by their mean and standard deviation respectively. Then, we perform minibatch gradient descent on the DDPO objective over the sample sequences (batch size of 32).

Unless otherwise specified, all RL models are trained for 1000 steps. We used the Adam optimizer, a learning rate of $10^{-5}$ and an effective batch size of 32 (again over 4 Nvidia A10 GPUs). As in Black et al. (2024), we used clipping to enforce a trust region for the policy update, with a clip value of 0.2.

**Protein folding**   During this phase of training, we sampled hundreds of thousands of sequences from the policy. Folding all of these sequences is computationally challenging, so we use ESM-Fold Lin et al. (2023) instead of AlphaFold2 to strike a balance between speed and accuracy. We leave it to future work to investigate more accurate folding algorithms Jumper et al. (2021); Wu et al. (2022).

We used the ESMFold implementation in the Huggingface Transformers library Research and wrapped it in an MLflow v2.3.6 framework. This was deployed on a Kubernetes cluster containing 20 Nvidia A10 GPUs, with an application load balancer to distribute traffic evenly. This architecture was critical to enable efficient on-policy training, as folding generated sequences is far more computationally-intensive than updating the policy.

## 4   RELATED WORKS

**Inverse folding models**   A majority of existing IF methods utilize transformers (GraphTrans Ingraham et al. (2019) and ESM-IF Hsu et al. (2022)) or graph neural networks in which nodes are amino acids, edges are defined between amino acids close together in the protein structure, and node and edge features are constructed from the protein structure backbone (ProteinMPNN Dauparas et al. (2022b), PiFold Gao et al. (2022), Grade-IF Yi et al. (2023a)). The learning objective of these methods can be discriminative Gao et al. (2022) or autoregressive Dauparas et al. (2022b); Hsu et al. (2022); Ingraham et al. (2019).

More recently, GradeIF Yi et al. (2023a) introduced a diffusion-based IF method. We note that the GradeIF results are competitive, but use solvent accessible surface area (SASA) features, which are strongly correlated with amino acid identity. These additional features are not typically part of the IF task specification and their effect is studied in Appendix A.1.

KWDesign Gao et al. (2024) fuses information from pre-trained protein structure and language models (GearNet Lopes & Costa (2013), ESM2Lin et al. (2022), and ESMIFHsu et al. (2022)) to improve amino acid representations and boost sequence recovery to 61%.

Discrete Flow Models Campbell et al. (2024) can also be applied to the IF setting, generalizing the discrete diffusion approach used by this work and others. Building on this, Discrete Guidance Nisonoff et al. (2024) guides the sampling trajectory towards high-quality samples. In the context of IF, this has been demonstrated to improve protein stability. While Discrete Guidance has not been applied to improve structural consistency, we note that it is a complementary method to ours.

**RL for biological sequence diffusion**   SEIKO Uehara et al. (2024) proposes a framework for online tuning of a diffusion model given a reward model and compares their method to DDPO and DPOK Fan et al. (2024). Among the evaluated problems is the design of green fluorescence protein (GFP) sequences. We note that the IF task differs from GFP design in that the former is a conditional generation task, and the primary goal is to generalize to conditions (i.e. structures) not observed during training.

## 5   EXPERIMENTS

In this section, we benchmark RL-DIF against SOTA models on different datasets and compare their foldable diversity and structural consistency (sc-TM).

### 5.1   BENCHMARKING DATASETS

We benchmarked on the following datasets:

- **CATH 4.2**: This is identical to the test hold-out of the data described in Section 3.2. We further use the partitioning of the test data by Ingraham et al. (2019) into a "short" subset of all proteins shorter than 100 residues (94 proteins); a "single" subset of all single chain proteins (102 proteins)

- **TS50 and TS500**: TS50 and TS500 Li et al. (2014) are curated lists of proteins of size 50 and 500 respectively from the PISCES server Wang & Dunbrack (2003).

- **CASP15**: The CASP15 dataset comprises of 45 protein structures used to assess the quality of forward-folding models as these structures are de-novo protein structures.

In Appendix A.2, we assess the structural and sequence similarity between proteins in these datasets and the CATH 4.2 training set. We find that TS50, TS500, and CASP15 have cross-split overlaps (as defined by SPECTRA Ektefaie et al. (2024)) ranging from 42-84%, indicating strong overlap with the training set. We therefore include these datasets for consistency with previous work, but focus the comparison on CATH 4.2

## 5.2 Model sampling strategies

To sample diverse sequences from IF models, different strategies can be employed, depending on how the model parameterizes $p(S|X)$:

- **Single-shot models**: These models parameterize $p(S|X)$ by factorizing the joint distribution into $\prod_{i=1}^{N} p(S[i]|X)$. This distribution can be reshaped by a temperature parameter. A temperature of 0 always picks the highest-probability AA, and higher temperatures encourage greater diversity.

- **Autoregressive models**: AR models return a probability distribution over possible amino acids for a given residue, conditioned on already-sampled residues. As in the single-shot case, we can reshape the conditional distributions with temperature.

  If an AR model is trained with randomly-permuted residues (instead of left-to-right), then we may also randomize the order in which we decode residues. This allows diverse sampling even at a temperature of 0.

- **Diffusion models**: Since a diffusion model iteratively maps a random vector to an amino acid sequence, we can introduce diversity by repeatedly sampling from a stationary distribution $p(S_T)$. We do not reshape the model-predicted posterior $p_\theta(S_{t-1}|S_t)$.

For the specific IF models evaluated in our study, we use the following settings:

- **ProteinMPNN**: An AR model, so we use temperature sampling with temperatures of 0.1, 0.2, and 0.3. We also sample at a temperature of 0 with random decoding order.

- **PiFold**: A single-shot model, so we use temperature sampling with values 0.1 and 0.2.

- **KWDesign**: Another single-shot model, so we use the same settings as PiFold.

- **ESMIF**: An AR model. We follow the authors' recommendation and sample with a temperature of 1.

- **DIF-Only and RL-DIF**: Diffusion models, so we sample from the uniform distribution $p(S_T)$ and iteratively denoise sequences using the model.

## 5.3 Benchmarking and performance of IF models trained on CATH4.2 benchmark

**Experimental Setup**   We compare RL-DIF to ProteinMPNN Dauparas et al. (2022b), PiFold Gao et al. (2022), and KWDesign Gao et al. (2024). We also evaluate our model after the diffusion pre-training phase, before any RL-optimization (DIF-Only). For each model and each benchmark dataset, we run the following pipeline:

1. For each protein structure in the dataset, sample 4 sequences from the model.

2. Among the 4 sampled sequences, compute the mean sequence recovery and sc-TM score.

3. Compute the foldable diversity of the 4 sequences, as defined in Equation 8, with $\text{TM}_{min} = 0.7$.

Model weights were not available for KWDesign for sampling, however they provided the probability distributions for CATH4.2 and CASP15, and we sampled from these to generate sequences.

**Results**   Among the evaluated models, our proposed methods (DIF-Only and RL-DIF) achieve or match the highest foldable diversity on all benchmarks in Table 1. In particular, we highlight the significantly-increased diversity (29% versus the next-best 23%) on the CATH-all dataset (which includes multi-protein complexes), with structural consistency only slightly degrading or improving.

Figure 2 demonstrates that the improvement in foldable diversity is minimally dependent on the choice of $\text{TM}_{min}$. Despite our methods not always achieving the highest sc-TM score, they offer the highest diversity across a range relevant sc-TM thresholds.

Although our models do not have high sequence recoveries, we call attention to the argument put forth in Section 3.1: sequence recovery prefers sequences very close to the naturally-observed sequence, which is a small slice of the accessible design space. We therefore focus our comparison on foldable diversity and structural consistency.

Across the various benchmarks, we observe a trend between DIF-Only and RL-DIF: RL-tuning consistently improves structural consistency, but frequently at a cost to diversity. This implies that the DDPO objective is being effectively optimized, but the entropy of the policy is decreasing during the second training phase. We leave it to future work to incorporate stronger exploration strategies, which may reduce this effect.

To demonstrate the effect of high foldable diversity, we present example protein sequences in Figure 1. All models inverse fold structurally self-consistent sequences, but with varying degrees of diversity.

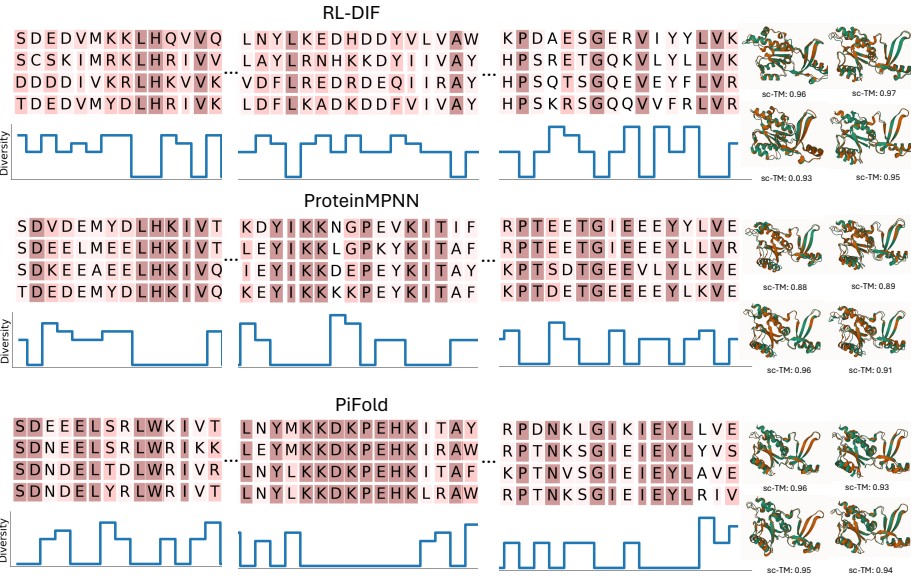

Figure 1: Samples from RL-DIF, ProteinMPNN, and PiFold, conditioned on the same protein backbone (PDB: 5FLM.E). The diversity of amino acids at each position is colour-coded, ranging from dark-red for "all identical" to no colour for "all different". While all models achieve high structural consistency (ESMFold-ed structures shown on the right), RL-DIF generates the most diverse set.

## 5.4 COMPARISON OF RLDIF PERFORMANCE TO ESM-IF

**Experimental Setup**   ESM-IF Hsu et al. (2022) is another protein inverse folding model that was trained on CATH 4.3 and 12 million synthetic data samples generated by AlphaFold 2 Jumper et al.

Table 1: Benchmarking results of ProteinMPNN, PiFold, KWDesign, DIF-Only, and RLDIF on CATH-single, CATH-short, CATH-all, TS50, TS500, and CASP15 reporting foldable diversity, sc-TM, and sequence recovery at various temperatures. T=X indicates temperature sampling occured at a temperature of X. RD indicates random decoding order. Best results for each dataset are bolded.

| Dataset | Model | Foldable Diversity↑ | sc-TM↑ | Sequence Recovery |
|---|---|---|---|---|
| CATH-short | ProteinMPNN (T=0, RD) | 9% | 0.55 | 31% |
| | ProteinMPNN (T=0.1) | 5% | 0.41 | 32% |
| | ProteinMPNN (T=0.2) | 0.2% | 0.19 | 22% |
| | ProteinMPNN (T=0.3) | 0% | 0.15 | 15% |
| | PiFold (T=0.1) | 6% | 0.44 | 35% |
| | PiFold (T=0.2) | 1% | 0.22 | 33% |
| | KWDesign (T=0.1) | **10%** | 0.53 | 40% |
| | KWDesign (T=0.2) | 4% | 0.54 | 31% |
| | DIF-Only | 8% | 0.42 | 34% |
| | RL-DIF | **10%** | 0.51 | 37% |
| CATH-single | ProteinMPNN (T=0, RD) | **17%** | 0.64 | 29% |
| | ProteinMPNN (T=0.1) | 11% | 0.46 | 30% |
| | ProteinMPNN (T=0.2) | 0% | 0.17 | 23% |
| | ProteinMPNN (T=0.3) | 0% | 0.14 | 14% |
| | PiFold (T=0.1) | 10% | 0.47 | 33% |
| | PiFold (T=0.2) | 0.1% | 0.20 | 24% |
| | KWDesign (T=0.1) | 14% | 0.61 | 42% |
| | KWDesign (T=0.2) | 8% | 0.55 | 32% |
| | DIF-Only | 13% | 0.48 | 32% |
| | RL-DIF | **17%** | 0.57 | 36% |
| CATH-all | ProteinMPNN (T=0, RD) | 20% | 0.80 | 41% |
| | ProteinMPNN (T=0.1) | 23% | 0.67 | 39% |
| | ProteinMPNN (T=0.2) | 3% | 0.30 | 28% |
| | ProteinMPNN (T=0.3) | 0.1% | 0.14 | 18% |
| | PiFold (T=0.1) | 23% | 0.72 | 44% |
| | PiFold (T=0.2) | 8% | 0.38 | 33% |
| | KWDesign (T=0.1) | 18% | 0.79 | 54% |
| | KWDesign (T=0.2) | 23% | 0.58 | 41% |
| | DIF-Only | **32%** | 0.72 | 40% |
| | RL-DIF | 29% | 0.78 | 44% |
| TS50 | ProteinMPNN (T=0, RD) | 21% | 0.84 | 48% |
| | ProteinMPNN (T=0.1) | 12% | 0.83 | 48% |
| | PiFold (T=0.1) | 0.5% | 0.86 | 57% |
| | KWDesign | N/A | N/A | N/A |
| | DIF-Only | **38%** | 0.77 | 45% |
| | RL-DIF | 30% | 0.83 | 50% |
| TS500 | ProteinMPNN (T=0, RD) | 23% | 0.87 | 52% |
| | ProteinMPNN (T=0.1) | 11% | 0.86 | 52% |
| | PiFold (T=0.1) | 0.4% | 0.87 | 59% |
| | KWDesign | N/A | N/A | N/A |
| | DIF-Only | **36%** | 0.83 | 48% |
| | RL-DIF | 28% | 0.84 | 54% |
| CASP15 | ProteinMPNN (T=0, RD) | 15% | 0.61 | 39% |
| | ProteinMPNN (T=0.1) | 15% | 0.55 | 39% |
| | PiFold (T=0.1) | 16% | 0.57 | 42% |
| | KWDesign (T=0.1) | 13% | 0.61 | 48% |
| | DIF-Only | **23%** | 0.55 | 38% |
| | RL-DIF | 18% | 0.60 | 42% |

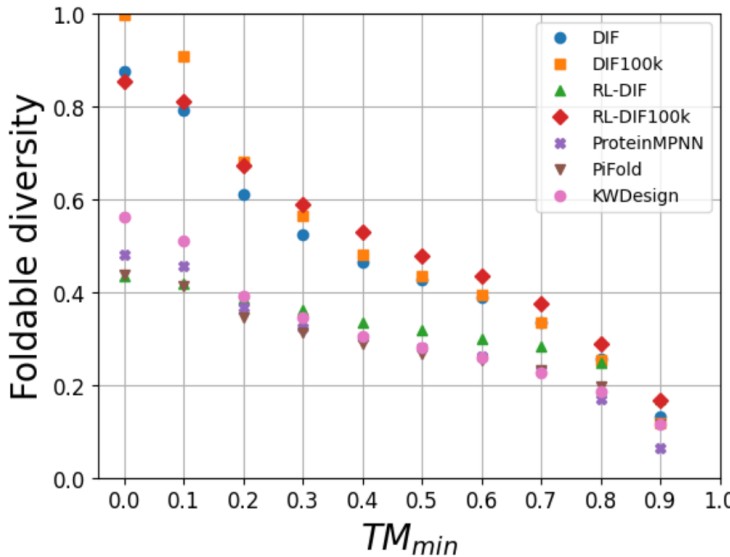

Figure 2: To evaluate the sensitivity of our analysis to the value of $TM_{min}$, we scan a range of values. We observe that DIF-Only and RL-DIF consistently perform the best, for thresholds $\geq 0.4$. Foldable diversity is computed across the "all" CATH 4.2 test split using the definition in Section 3.1.

(2021). While a large database of AlphaFold-ed structures is publicly available (EMBL-EBI), the ESM-IF paper did not release the IDs of the proteins that were trained on. These IDs are essential in order to replicate their work and ensure no overlap between the training and benchmark datasets.

To attempt to replicate their work, we followed a similar strategy of curating a larger pre-training dataset orthogonal to all benchmarking datasets we consider. This was done using Foldseek van Kempen et al. (2024) to ensure no structure or sequence overlap, resulting in a dataset of 100,000 structures from the AlphaFold database. We then retrained RL-DIF on this larger pretraining dataset, followed by the same 1000 steps of RL optimization. We refer to these models as DIF-Only-100K and RL-DIF-100K. In order to facilitate future work we release the IDs of the entries used to train RL-DIF-100K.

**Results** We found that, as compared to RL-DIF, RL-DIF-100K improves foldable diversity across all datasets. As demonstrated in Table 2, DIF-Only-100K and RL-DIF-100K approach (and in the case of TS50, surpass) ESM-IF's performance, despite using 70 times less data and 30 times fewer parameters.

Given the increased performance from expanding the pre-training dataset from 18K (CATH training set) to 100K, we leave it to future work to explore the effect of further dataset scaling. Ideally, we would have liked to extend the comparison to use 12M structures as done by the ESM-IF authors, but found this computationally intractable with our resources (in particular, Foldseek has linear memory complexity). In Figure S4, we show all considered IF models, comparing their foldable diversity and amount of training data.

## 5.5 THE EFFECT OF REINFORCEMENT LEARNING ON DIF-ONLY PERFORMANCE

To investigate the effect of reinforcement learning on DIF-Only performance we ran our RL for 4000 steps, taking the model at every 1000 steps and benchmarking its performance for all metrics. We found average sc-TM had the largest improvement in the first 1000 steps of RL before stagnating after step 2000 (Table 3). Foldable diversity continued to decrease with modest gains in structural consistency. These results support our decision to run RL for 1000 steps.

Table 2: Benchmarking results of ESM-IF, DIF-Only-100k, and RL-DIF-100K models on various datasets reporting foldable diversity, sc-TM, and sequence recovery. * indicates both results are statistically equivalent (p-value > 0.05)

| Dataset | Model | Foldable Diversity↑ | sc-TM↑ | Sequence Recovery |
|---|---|---|---|---|
| CATH 4.2 | ESM-IF | **37%** | 0.77 | 42% |
| | DIF-Only | 32% | 0.72 | 40% |
| | RL-DIF | 29% | 0.78 | 45% |
| | DIF-Only-100K | 33% | 0.68 | 34% |
| | RL-DIF-100K | 34% | 0.76 | 38% |
| TS50 | ESM-IF | **43%** | 0.83 | 47% |
| | DIF-Only | 38% | 0.77 | 45% |
| | RL-DIF | 30% | 0.83 | 50% |
| | DIF-Only-100K | 39% | 0.71 | 38% |
| | RL-DIF-100K | 39% | 0.80 | 42% |
| TS500 | ESM-IF | 38% | 0.86 | 51% |
| | DIF-Only | 36% | 0.83 | 48% |
| | RL-DIF | 28% | 0.84 | 54% |
| | DIF-Only-100K | **40%** | 0.79 | 41% |
| | RL-DIF-100K | 36% | 0.84 | 45% |
| CASP15 | ESM-IF | **24%*** | 0.59 | 38% |
| | DIF-Only | 23% | 0.55 | 38% |
| | RL-DIF | 18% | 0.60 | 42% |
| | DIF-Only-100K | 18% | 0.51 | 34% |
| | RL-DIF-100K | 21%* | 0.59 | 39% |

Table 3: Effect of RL-tuning of DIF-Only-100K, as measured by performance on the TS50 dataset.

| RL Steps | Foldable Diversity↑ | sc-TM↑ | Sequence Recovery |
|---|---|---|---|
| 0 (DIF-Only-100K) | 39% | 0.71 | 38% |
| 1000 (RL-DIF-100K) | 39% | 0.80 | 42% |
| 2000 | 35% | 0.81 | 43% |
| 3000 | 34% | 0.82 | 43% |
| 4000 | 32% | 0.81 | 44% |

## 6 CONCLUSION

Although sequence recovery, sequence diversity, and structural consistency are commonly used to evaluate protein IF models, here we demonstrate that those metrics alone do not necessarily capture the ability of the model to generate multiple sequences that fold into the desired structure. For many practical applications, foldable diversity is a useful metric since it gives users multiple "shots-on-goal" for filtering or optimizing designs on criteria beyond structural consistency. We also present new inverse folding models: DIF-Only and RL-DIF. We find that among evaluated methods, these models achieves highest foldable diversity and sc-TM in the CATH4.2, CASP15, TS50, and TS500 datasets.

Nonetheless, we note some limitations of RL-DIF. First is the exclusive use of ESMFold as the folding model. Alternative models such as AlphaFold2 Jumper et al. (2021), OpenFold Ahdritz et al. (2024), or OmegaFold Wu et al. (2022) have demonstrated superior folding accuracy and could potentially enhance our results. An ensemble of these models could also be employed to leverage the specific strengths and mitigate the weaknesses of each individual model. Additionally, instead of limiting our sampling to four iterations, we could continuously sample until achieving four samples with a sc-TM greater than 0.7. While this approach would likely improve model accuracy, it also significantly increases computational cost. Consequently, we imposed a limit of four samples to

manage computational resources. Finally, we could leverage entropy bonus methods used in RL to further increase sequence diversity and exploration of the policy.

## CODE AVAILABILITY

RL-DIF model weights and sampling code is currently available at `https://github.com/flagshippioneering/pi-rldif`.

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

# A APPENDIX

## A.1 GRADEIF RESULTS

When exploring the performance of GradeIF we found the solvent accessible surface area (SASA) feature is calculated utilizing side chain information. When plotting the SASA values per amino acid type, there is a clear separation between some amino acids which can inflate model performance (Figure S1). When we retrained GradeIF with recommended hyperparameters without (Figure S2a) and with SASA (Figure S2b) we noticed a 23% decrease in performance. With this in mind, we decided to pursue utilizing PiFold Gao et al. (2022) as the underlying architecture for the categorical diffusion.

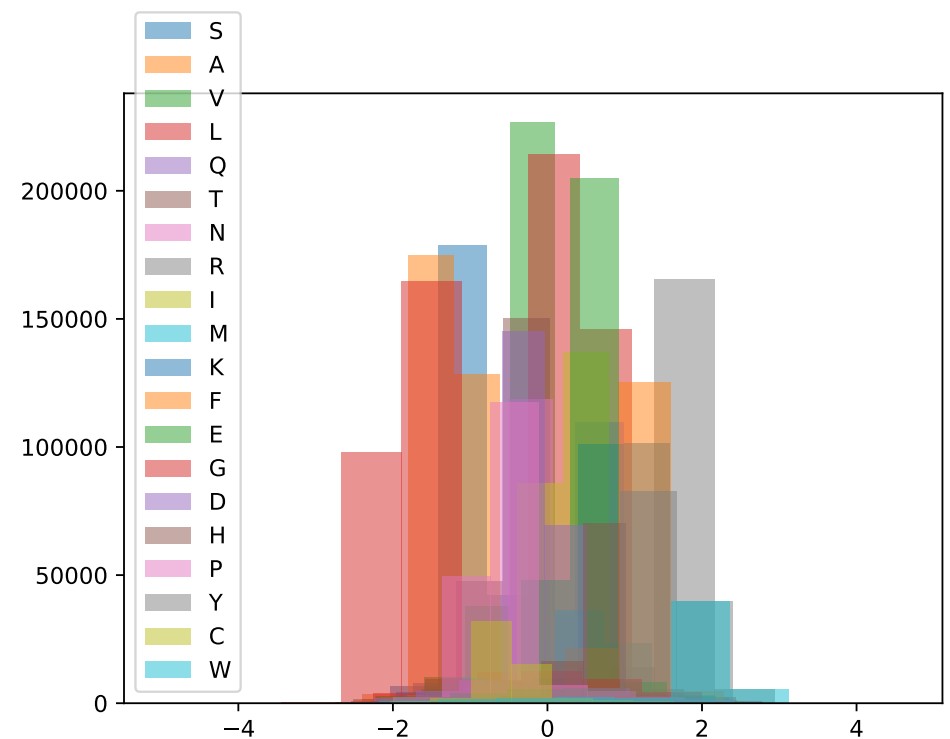

Figure S1: SASA feature value for every amino acid in CATH4.2 training set proteins.

## A.2 SPECTRA AND FOLDSEEK

To understand how model performance changes as a function of test protein similarity to the CATH4.2 training set, we utilize the spectral framework of model evaluation (SPECTRA) Ektefaie et al. (2024). We utilized Foldseek van Kempen et al. (2024) to determine if two proteins were similar. Two proteins are similar if e-value returned by Foldseek is greater than 1e-3 or the sequence similarity is greater than 0.3. Given two datasets, cross-split overlap is defined as the proportion of proteins that are similar.

```
#Given a directory of pdbs to create a pdb database to scan versus the
    CATH4.2 train set
foldseek createdb <path pdb directory> <database name>
#Given a pdb database scan relative to the CATH4.2 train set (CASP15 used
    for example here) to find similar structures/sequences
foldseek search casp15 cath_train aln_casp15 ./res_casp15/ -s 7.5
#Convert output of search to final alignment tab output
foldseek convertalis casp15 cath_train aln_casp15 casp15.m8
```

Listing 1: Commands used to run Foldseek

Using this procedure we evaluated the structural sequence similarity between each benchmarking set and the CATH 4.2 training set. We found TS50, TS500, and CASP15 datasets had high levels of similarity to the CATH 4.2 train dataset (Table S1).

## A.3 FOLDING SUCCESS AND DIVERSITY AMONG FOLDED SEQUENCES

Foldable diversity (FD) incorporates two measurements (1) the proportion of proteins for which sequences can be generated that fold into the protein structure (folding success) and (2) the diversity

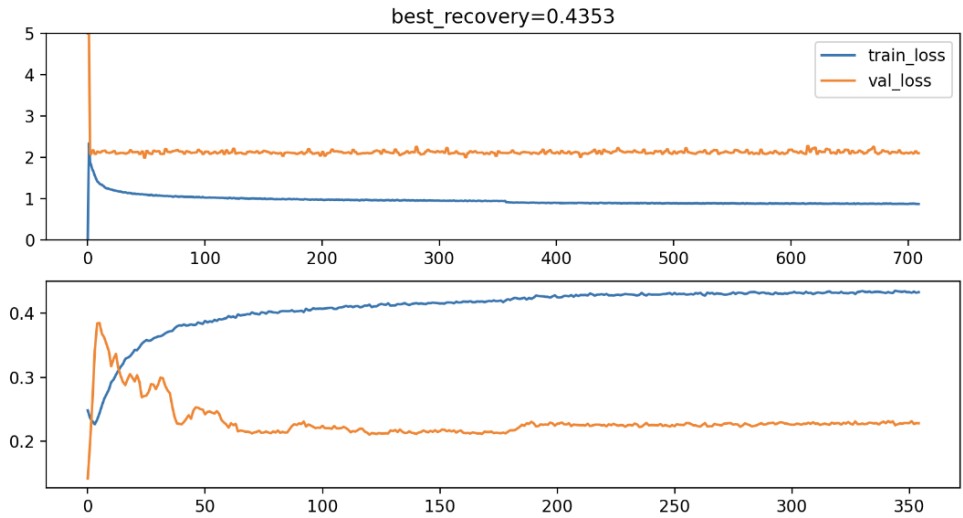

(a) GradeIF performance before removal of surface features. The top half of the plot shows train (blue) and val (orange) loss. The bottom half of the plot shows test perplexity (orange) and sequence recovery (blue).

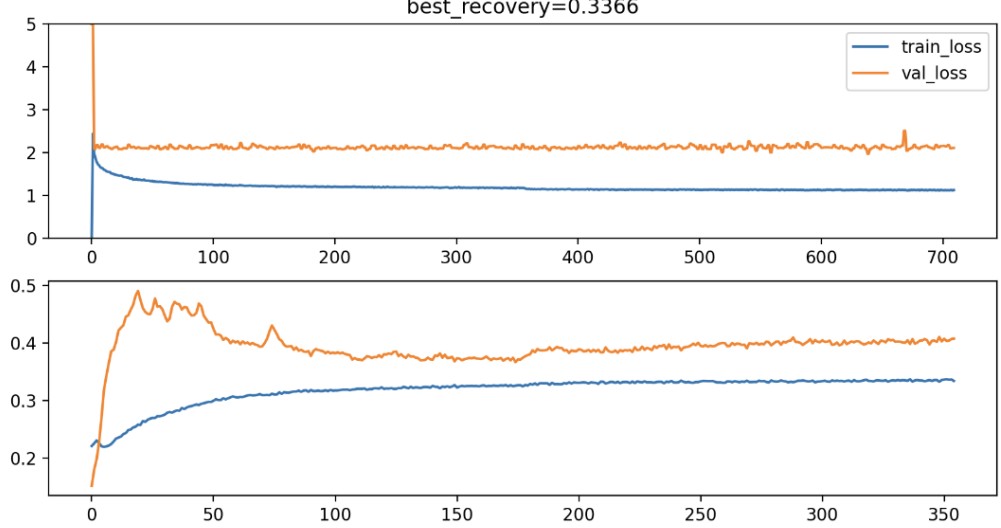

(b) GradeIF performance after removal of surface features. The top half of the plot shows train (blue) and val (orange) loss. The bottom half of the plot shows test perplexity (orange) and sequence recovery (blue).

Figure S2: GradeIF performance before and after removal of surface features.

Table S1: CATH 4.2 all, CATH 4.2 single, CATH 4.2 short, TS50, TS500, and CASP15 test set cross-split overlap with CATH 4.2 train set. Cross-split overlap is defined as the proportion of proteins in dataset that had similar sequence or structure to a protein in the CATH 4.2 train set (Appendix A.2).

| Model | Cross-split overlap |
|---|---|
| CATH 4.2 All | 18% (201/1120) |
| CATH 4.2 Single | 18% (19/103) |
| CATH 4.2 Short | 14% (13/94) |
| TS50 | 80% (40/50) |
| TS500 | 84% (420/500) |
| CASP15 | 42% (19/45) |

of the sequences that are able to fold into the protein structure. Foldable diversity decreases if either of these measures decrease. We can quantify (2) by introducing folded diversity (FdD): the same as Equation 8, but restricting to the subset of sequences that fold correctly. To de-convolve the relationships between FD, FdD, and folding success, we report all three metrics (Table S2).

Our findings indicate that increasing sampling temperature leads to an expected trade-off: FdD rises at the expense of folding success. That is, diversity can be increased, but only by rejecting a large fraction of sampled sequences. While, DIF-Only and RL-DIF also demonstrate a tension between FdD and folding success, they achieve competitive or improved FD scores.

Across most datasets, RL-DIF shows improved folding success, aligning with our expectations since sc-TM was used as the reward function during reinforcement learning. Table S4 reveals a significant increase in folding success within the first 1000 steps of RL-DIF training, followed by a decline and stabilization. This trend is also evident in models trained with more data (Table S3), where RL-DIF-100K consistently surpasses DIF-Only-100K in folding success.

A trade-off between folded diversity and folding success is documented between DIF-Only-100K and RL-DIF-100K, with DIF-Only-100K achieving higher folded diversity, even surpassing ESMIF across tested datasets, but at the cost of reduced folding success. Ultimately, the choice of model depends on the specific application, so we report both to guide selection.

### A.4    THE EFFECT OF EXTRA SAMPLING ON FOLDABLE DIVERSITY

It is computationally infeasible to sample and fold hundreds of sequences for benchmarking purposes. However, to investigate the impact of sampling on foldable diversity, we sampled 100 sequences from each of three PDB structures: 8JZ1, 8QOC, and 1ANF. We selected these three PDB structures for their structural diversity and limited our choice to three due to the high computational cost of folding hundreds of sequences. We first calculated the foldable diversity for the entire set of 100 sequences. Then, we randomly selected 100,000 groups, each consisting of n sequences (with n being 2, 4, 8, 10, and 20) from these 100 sequences. For each group, we calculated its foldable diversity and determined the mean absolute difference between these groups and the full set of 100 sequences. (Figure S6). We then compare these mean differences to the minimum margin by which RLDIF outperforms the next best model (7%).

The maximum mean difference sampling at 4 sequences was 3.6% less than the minimum 7% margin observed in our results. This implies that if we sampled more than 4 sequences our results would not change, as we observe the mean difference only drops as the number of samples increases (Figure S6). However, it is important to note the margin for CATH-single and CATH-short in (Table 1) is 0% and thus falls within this margin of error. This would imply more sampling would be necessary to effectively discern which model outperforms the other for these datasets, but for all other benchmarks our results remain valid.

### A.5    SUPPLEMENTARY TABLES AND FIGURES

Table S2: Same as Table 1, except reporting diversity for sequences with an sc-TM over 0.7 (folded diversity) and the proportion of samples with generated sequences achieving an sc-TM over 0.7 (folding success). Foldable diversity encompasses both folded diversity and folding success: a model achieves higher foldable diversity by effectively balancing the observed trade-off between the other two metrics.

| Dataset | Model | Foldable Diversity ↑ | Folded Diversity ↑ | Folding Success ↑ |
|---|---|---|---|---|
| CATH-short | ProteinMPNN (T=0, RD) | 9% | 32% | 50% |
| | ProteinMPNN (T=0.1) | 5% | 45% | 29% |
| | ProteinMPNN (T=0.2) | 0.2% | 67% | 4% |
| | ProteinMPNN (T=0.3) | 0% | 0% | 0% |
| | PiFold (T=0.1) | 6% | 44% | 35% |
| | PiFold (T=0.2) | 1% | 66% | 4% |
| | KWDesign (T=0.1) | 10% | 14% | 42% |
| | KWDesign (T=0.2) | 4% | 54% | 24% |
| | DIF-Only | 8% | 50% | 30% |
| | RL-DIF | 10% | 42% | 48% |
| CATH-single | ProteinMPNN (T=0, RD) | 17% | 34% | 66% |
| | ProteinMPNN (T=0.1) | 11% | 46% | 39% |
| | ProteinMPNN (T=0.2) | 0% | 0% | 2% |
| | ProteinMPNN (T=0.3) | 0% | 0% | 0% |
| | PiFold (T=0.1) | 10% | 42% | 45% |
| | PiFold (T=0.2) | 0.1% | 64% | 3% |
| | KWDesign (T=0.1) | 14% | 32% | 60% |
| | KWDesign (T=0.2) | 8% | 53% | 38% |
| | DIF-Only | 13% | 49% | 41% |
| | RL-DIF | 17% | 43% | 60% |
| CATH-all | ProteinMPNN (T=0, RD) | 20% | 27% | 83% |
| | ProteinMPNN (T=0.1) | 23% | 42% | 71% |
| | ProteinMPNN (T=0.2) | 3% | 67% | 25% |
| | ProteinMPNN (T=0.3) | 0.1% | 0.1% | 0% |
| | PiFold (T=0.1) | 23% | 38% | 75% |
| | PiFold (T=0.2) | 8% | 63% | 40% |
| | KWDesign (T=0.1) | 18% | 26% | 81% |
| | KWDesign (T=0.2) | 23% | 54% | 65% |
| | DIF-Only | 32% | 52% | 76% |
| | RL-DIF | 29% | 28% | 83% |
| TS50 | ProteinMPNN (T=0, RD) | 21% | 26% | 88% |
| | ProteinMPNN (T=0.1) | 12% | 15% | 86% |
| | PiFold (T=0.1) | 0.5% | 0.5% | 86% |
| | KWDesign | N/A | N/A | N/A |
| | DIF-Only | 38% | 50% | 86% |
| | RL-DIF | 30% | 38% | 86% |
| TS500 | ProteinMPNN (T=0, RD) | 23% | 24% | 90% |
| | ProteinMPNN (T=0.1) | 11% | 13% | 89% |
| | PiFold (T=0.1) | 0.4% | 0.5% | 89% |
| | KWDesign | N/A | N/A | N/A |
| | DIF-Only | 36% | 46% | 88% |
| | RL-DIF | 28% | 35% | 90 % |
| CASP15 | ProteinMPNN (T=0, RD) | 15% | 28% | 64% |
| | ProteinMPNN (T=0.1) | 15% | 40% | 56% |
| | PiFold (T=0.1) | 16% | 39% | 58% |
| | KWDesign (T=0.1) | 13% | 28% | 62% |
| | DIF-Only | 23% | 55% | 58% |
| | RL-DIF | 18% | 43% | 58% |

Table S3: Same as Table 2, except comparing foldable diversity (FD), foldable diversity (FdD) and folding success.

| Dataset | Model | Foldable Diversity ↑ | Folded Diversity ↑ | Folding Success ↑ |
|---|---|---|---|---|
| CATH 4.2 | ESM-IF | 37% | 53% | 81% |
| | DIF-Only | 32% | 52% | 76% |
| | RL-DIF | 29% | 40% | 83% |
| | DIF-Only-100K | 33% | 60% | 73% |
| | RL-DIF-100K | 34% | 50% | 80% |
| TS50 | ESM-IF | 43% | 52% | 88% |
| | DIF-Only | 38% | 50% | 86% |
| | RL-DIF | 30% | 38% | 86% |
| | DIF-Only-100K | 39% | 58% | 80% |
| | RL-DIF-100K | 39% | 48% | 82% |
| TS500 | ESM-IF | 38% | 47% | 89% |
| | DIF-Only | 36% | 46% | 88% |
| | RL-DIF | 28% | 35% | 90% |
| | DIF-Only-100K | 40% | 56% | 85% |
| | RL-DIF-100K | 36% | 45% | 88% |
| CASP15 | ESM-IF | 24% | 56% | 60% |
| | DIF-Only | 23% | 55% | 58% |
| | RL-DIF | 18% | 43% | 58% |
| | DIF-Only-100K | 18% | 63% | 51% |
| | RL-DIF-100K | 21% | 49% | 60% |

Table S4: Same as Table 3, except comparing foldable diversity (FD), foldable diversity (FdD) and folding success.

| RL Steps | Foldable Diversity ↑ | Folded Diversity ↑ | Folding Success ↑ |
|---|---|---|---|
| 0 (DIF-Only-100K) | 39% | 58% | 80% |
| 1000 (RL-DIF-100K) | 39% | 38% | 86% |
| 2000 | 35% | 45% | 84% |
| 3000 | 34% | 42% | 84% |
| 4000 | 32% | 39% | 84% |

Table S5: Benchmarking results of ESMIF and augmented DIF-Only and RLDIF models on CATH 4.2 test hold-out structures reporting foldable diversity, sc-TM, and sequence recovery.

| Model | Foldable Diversity↑ | | | sc-TM↑ | | | Sequence Recovery↑ | | |
|---|---|---|---|---|---|---|---|---|---|
| | Short | Single | All | Short | Single | All | Short | Single | All |
| ESMIF | **13%** | **22%** | **37%** | 0.50 | 0.52 | 0.77 | 26% | 24% | 42% |
| DIF-Only+DB (100K) | 9% | 11% | 33% | 0.40 | 0.44 | 0.68 | 27% | 26% | 34% |
| RLDIF+DB (100K) | 12% | 17% | 34% | 0.48 | 0.54 | 0.76 | 30% | 28% | 38% |

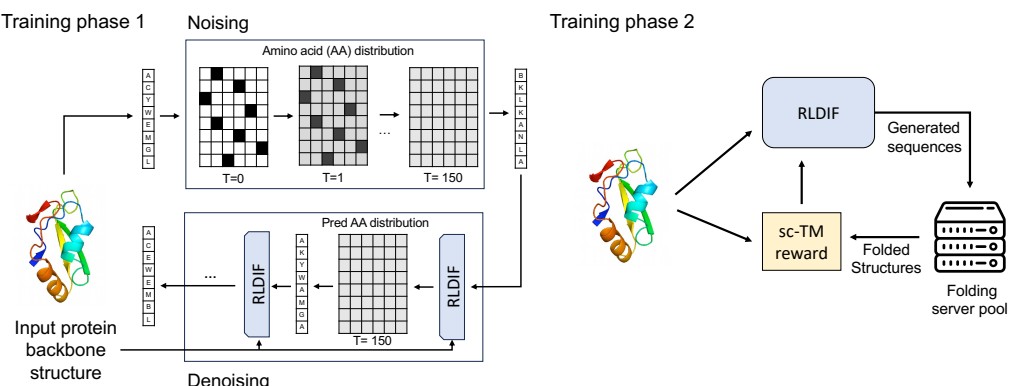

Figure S3: The overall framework for RL-DIF. Training phase 1: RL-DIF is pretrained to generate amino acid sequences conditioned on protein backbone structures using a categorical diffusion objective. Training phase 2: RL-DIF is refined to maximize the expected structural consistency of its generated sequences.

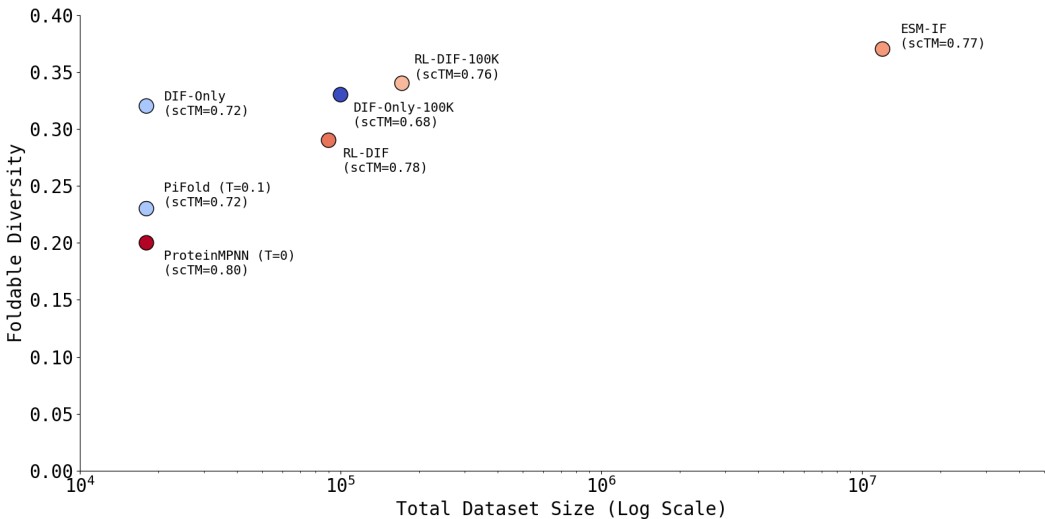

Figure S4: The effect of data scale on IF model performance. In general, we observe a positive relationship between additional data and foldable diversity, though this is not readily observable for structural consistency (indicated by color and labels). Note that, in the case of RL-tuned models, we include the sequences sampled during on-policy training in the dataset size.

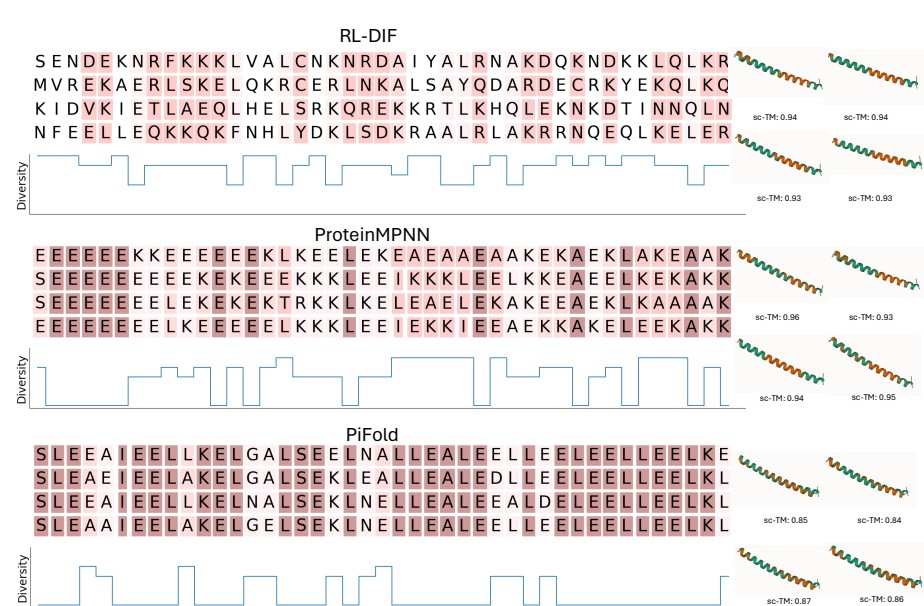

Figure S5: Sequence designs from 3 different IF models for a short single alpha-helix peptide structure (PDB: 2X7R.B)

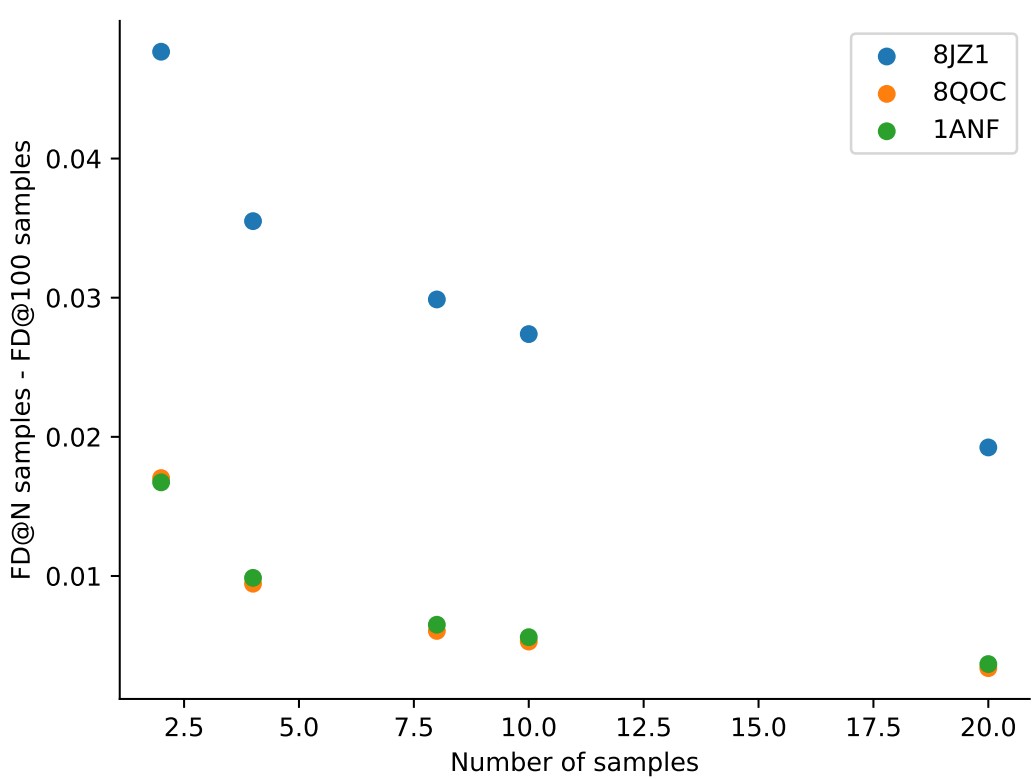

Figure S6: Mean absolute difference in foldable diversity between sampling n samples (x-axis) and 100 samples, sampled 100,000 times for PDB structures 8JZ1, 8QOC, and 1ANF.

