# OpenReview forum: "Reinforcement learning on structure-conditioned categorical diffusion for protein inverse folding"
_ICLR.cc/2025/Conference — Submitted to ICLR 2025_

### Official Review · Reviewer_6Vm7 · 2024-10-30

**Soundness:** 2
**Presentation:** 3
**Contribution:** 2
**Rating:** 6
**Confidence:** 5

**Summary:**

The paper presents RL-DIF, a discrete diffusion model trained for protein inverse folding and further fine-tuned on structural consistency. RL-DIF achieves comparable sequence recovery and structural consistency to existing methods, while having a better foldable diversity, beneficial to downstream optimizations.

**Strengths:**

- The proposed method combines recent advances in discrete diffusion and reinforcement learning alignment methods to improve the sequence diversity of protein inverse folding, an essential task in protein biology.
- The paper is easy to follow, and the experiments on various protein datasets show the improvement in diversity with the proposed method.

**Weaknesses:**

- The novelty of RL-DIF is limited. Several existing papers utilize discrete diffusion models for protein inverse folding, for example, [1] and [2]. The idea of using structural consistency to fine-tune the inverse folding model has been explored in ESM3[3], although with language model as the base model.
- The motivation for fine-tuning the model with structural consistency to improve the sequence diversity is not very clear. This is also reflected in the experiments in Table 1, where DIF-Only achieves higher diversity in most datasets except CATH-short and CATH-single.
- The Foldable Diversity is only calculated based on 4 generated sequences for each protein structure, which can have high variance. It is better to use a higher number of generated sequences, and report the standard deviation of the results across random seeds. Besides, it is also helpful to provide the ratio of the generated sequences that satisfy the sc-TM score constraint to give some idea of to what degree the Foldable Diversity is affected by the structural consistency.
- The model is compared to only non-diffusion based inverse folding baseline methods. A comparison with diffusion-based methods [1,2] is missing.
- Hamming distance is used to measure sequence diversity. However, it may not be able to capture the high-level diversity between sequences, which is more related to protein functions and more useful in practice. Thus, the distance calculated in a pretrained embedding space (eg. ESM embedding) could be better.

[1] Generative Flows on Discrete State-Spaces: Enabling Multimodal Flows with Applications to Protein Co-Design. ICML 2024.

[2] Unlocking Guidance for Discrete State-Space Diffusion and Flow Models. arXiv 2024.

[3] Simulating 500 million years of evolution with a language model. bioRxiv 2024.

**Questions:**

- How are the temperature values for the pretrained models selected? In Table 1, T=0.1 is much better than T=0.2 and T=0.3 for all baselines. How do they perform with a smaller T, eg. T=0.05 or T=0.01?
- The reviewer understands that it is computationally expensive to train RL-DIF on the same amount of data as ESM-IF. For a rough comparison, how is the performance of ESM-IF trained on the same dataset as RL-DIF-100K?

---

> ### Author Response · Authors · 2024-11-22
> **Authors' response**
>
> >The novelty of RL-DIF is limited. [...]
>
> We address the comparison to discrete diffusion/flow models below, but would like to motivate the contributions of our work here:
>
> (1) While this work does combine a number of prior methods, structural consistency was a difficult reward to optimize. In general, the methods we were building on top of did not work out of the box. To summarize the problem-specific optimizations:
>  * Removing SASA features hampered the GradeIF architecture (see Appendix A1). We came up with a variant of the PiFold that supported discrete diffusion to mitigate this.
>  * Our experiments showed that the D3PM hybrid loss out-performed GradeIF’s cross-entropy loss
>  * We found the optimization to be sensitive to DDPO hyperparameters and reward standardization, requiring careful tuning
>  * ESM3 does use structural consistency to improve an IF model, but we note some methodological differences (IRPO vs DDPO) and difficulty comparing performance (closed training set may have overlap with IF benchmark datasets).
>
> (2) We believe foldable diversity is an important contribution. IF methods have been primarily benchmarked using sequence recovery, sequence diversity, and structural consistency. Our work shows that these metrics alone do not fully capture the main purpose of an inverse folding model: sampling diverse sequences that fold into a target structure.
>
>
> >The motivation for fine-tuning the model with structural consistency to improve the sequence diversity is not very clear.
>
> We were motivated to fine-tune the model with structural consistency to improve structure quality, since continual training on sequence recovery plateaus in sc-TM and decreases diversity. We find DIF-Only achieves high diversity but poor sc-TM; RL-DIF then improves sc-TM significantly while maintaining competitive diversity (Table S2, S4).
>
> >The Foldable Diversity is only calculated based on 4 generated sequences for each protein structure, which can have high variance.
>
> We chose 4 to strike a balance between a precise measurement and minimizing computational resources In appendix A4, we study the impact of 4 vs 100 sequences and empirically observe a difference of at most 3%. Comparing to Table 1, we conclude that our improvements are outside of this error bar (i.e. on CATH-all, RL-DIF has a margin of 6%; DIF-Only by 9%).
>
> >Besides, it is also helpful to provide the ratio of the generated sequences that satisfy the sc-TM score constraint
>
> We have added Appendix A3, referring to the ratio you propose as “folding success”. We observe that a low folding success always leads to a low FD. However, high folding success does not guarantee high FD, as the sequences may not be diverse. It is this relationship that motivated our definition of FD.
>
> >A comparison with diffusion-based methods [1,2] is missing.
>
> Thank you for pointing out our oversight of these papers - we have added them to the Related Works section of the manuscript. However, we do not benchmark against either:
> * As there are many IF methods, we focus on SOTA models at time of writing; Campbell et al reports sc-RMSD greater than ProteinMPNN on CATH.
> * Nisinoff et al focused their stability optimization on specific proteins (instead of a full IF benchmark dataset), so we did not benchmark their model.
>
> >Hamming distance is used to measure sequence diversity. [...]
>
> In the IF setting, we are typically designing protein sequences for a very specific function (making the approximation that structure determines function). Therefore, significant functional diversity is not desirable.
>
> Sequence diversity is desirable because designs may fail evaluation for a priori unknown reasons - e.g. properties like synthesizability. Distance in ESM embedding space will be dominated by evolutionary and functional variation, not such properties. We prefer Hamming distance as it is interpretable and commonly used in prior IF work.
>
> Questions:
> > How are the temperature values for the pretrained models selected? ... How do they perform with a smaller T, eg. T=0.05 or T=0.01?
>
> We evaluated ProteinMPNN at a temperature of 0.05 and 0.01 on TS50; FD decreases to 5% and 1% respectively. The temperatures in the manuscript are those previously found to achieve best performance in ProteinMPNN, PiFold, and KWDesign [1].
>
> [1] https://openreview.net/forum?id=mpqMVWgqjn; Figure 6
>
> > For a rough comparison, how is the performance of ESM-IF trained on the same dataset as RL-DIF-100K?
>
> ESM-IF authors have not released their training code, so we are not able to reproduce their work.
>
> Instead, we trained a DIF-Only-250k on 250k structures, resulting in TS50 FD comparable to ESM-IF (see table below), with 18x fewer parameters and 48x less data. It remains for us to benchmark on CATH and TS500, and repeat RL tuning.
>
> | Model | FD | sc-TM |
> | --------|----| -- |
> | ESMIF | 43% | 0.83 |
> | DIF-Only-100k | 39% | 0.71 |
> | RL-DIF-100k | 39% | 0.80 |
> | DIF-Only-250k | 43% | 0.75 |

---

> > ### Author Response · Authors · 2024-11-25
> > **Reviewer 6Vm7 Followup**
> >
> > We would like to ask the reviewer if our response has resolved their outstanding comments. Given the deadline closing tommorow, we want to see if we could provide any more analysis or clarification to demonstrate the strength of our work.

---

> ### Comment · Reviewer_6Vm7 · 2024-11-28
>
> Thanks the authors for the detailed clarifications. However, my main concerns about the motivations and novelty remain: (1) While acknowledging the difference between this paper and existing works mentioned by the authors, the modifications still seem marginal to me. (2) The motivation to fine-tune the model with structural consistency is to improve structure quality, which is not directly/necessarily related to structural diversity. In terms of a better diversity-quality boundary, instead of stating that structural consistency fine-tuning improves diversity, it is more reasonable to view it as improved IF quality given a diversity level. (3) A comparison to diffusion-based models is necessary because they are also generative models with similar types of random sampling processes, which can naturally boost diversity compared to methods like ProteinMPNN. Also, in Table 3 of [2], FMIF outperforms ProteinMPNN on its PDB test set (which is a full IF benchmark dataset).
>
> Therefore, I will maintain my score for these reasons.

---

> > ### Author Response · Authors · 2024-12-01
> > **Response to Reviewer 6Vm7**
> >
> > We thank the reviewer for their continued engagement and suggestions.
> >
> > To respond to the points:
> >
> > (1) Regarding methodological innovations, we believe our empirical results underscore the value of our work:
> >
> > 1. RDIF/DIF-Only achieves state-of-the-art results across all evaluated datasets.
> > 2. Scaling RLDIF/DIF-Only delivers competitive performance with ESMIF, using 18x fewer parameters and 48x times less data.
> >
> > Additionally, we release the pretraining data to enable broader use and further development of inverse folding models—an openness not matched by ESMIF, which did not release its training code and dataset. This is in addition to the other methodological modifications mentioned earlier.
> >
> > (2) We apologize for the confusion. No where in the paper do we state that refining on structural consistency improves diversity. In fact we mention that refining on structural consistency decreases diversity. For example in line 342:   “RL-tuning consistently improves structural consistency, but frequently at a cost to diversity.”
> >
> > If there is somewhere where the reviewer feels like we implied this we ask the reviewer to point it out so we can remedy the presentation.
> >
> > The reviewer points out that it is “more reasonable to view it as improved IF quality given a diversity level.” We agree it was never our motivation to fine-tune on structural consistency to improve diversity, in the introduction we state we fine-tune the diffusion model “to maximize the expected structural consistency of designed sequences. (Line 57-58).” When we present our results, readers can observe that RLDIF improves IF quality given a diversity level. We will add a few sentences to the results section to emphasize this point.
> >
> > (3) Table 3 shows FMIF outperforms ProteinMPNN in sequence recovery which we established in this work does not correlate with structural consistency or foldable diversity. Nevertheless, the paper “Unlocking Guidance for Discrete State-Space Diffusion and Flow Models” has this GitHub “https://github.com/andrew-cr/discrete_flow_models/?tab=readme-ov-file” which then points to this GitHub “https://github.com/jasonkyuyim/multiflow” for the protein experiments which is the GitHub for the second paper the reviewer mentioned “Generative Flows on Discrete State-Spaces: Enabling Multimodal Flows with Applications to Protein Co-Design.” We followed the instructions the authors of the paper provided but all generated sequences are a series of As which is an active GitHub issue that the authors have not resolved. We already attempted to benchmark with GradeIF but ran into problems that disqualify it from consideration, as outlined in the Appendix.
> >
> > We ask the reviewer to reconsider their score in light of these responses.

---

> ### Comment · Reviewer_6Vm7 · 2024-12-02
> **Response to authors**
>
> Thank the authors for the further clarifications about the contribution and motivation of their work and for attempting diffusion-based models. In light of this, I increase my score. I would like the authors to add these clarifications in the revised manuscript. Regarding the diffusion-based models, I understand the time limitation of the rebuttal period. I would expect an inverse folding model to work well if the authors follow the steps in Section F.4.1 in [2] and encourage the authors to have a try.

---

### Official Review · Reviewer_R2ES · 2024-11-03

**Soundness:** 3
**Presentation:** 3
**Contribution:** 3
**Rating:** 5
**Confidence:** 4

**Summary:**

This paper introduces RL-DIF, a novel reinforcement learning-driven model for protein inverse folding, addressing the need for generating diverse ensembles of amino acid sequences that fold into specific, target 3D protein structures via inverse folding. Authors introduce a framework they call RL-DIF that applies a categorical diffusion approach pre-trained on sequence recovery, followed by reinforcement learning (RL) fine-tuning to optimize structural consistency. This two-phase training strategy enables RL-DIF to improve "foldable diversity"—the diversity of sequences that maintain the correct fold—while retaining sequence recovery and structural accuracy. In tests on CATH 4.2 and other datasets, RL-DIF achieved up to 29% foldable diversity, outperforming prior models, which peaked at 23%.
Key contributions of the paper include:
RL-DIF Model: A diffusion-based model refined with denoising diffusion policy optimization to balance diversity and structural fidelity.
Benchmarking Results: Demonstrating RL-DIF’s superior foldable diversity on multiple datasets.
Methodological Advancements: Introducing foldable diversity as a new metric to assess inverse folding model quality.
The RL-DIF model code and PyTorch weights are available in HF.

**Strengths:**

Paper tables an important problem in inverse folding with an interesting solution in the combination of diffusion model and RL. It's an exciting contribution that could open up a new set of metrics for inverse folding models. The diversity of inverse folding models is a known problem in practice, but few direct attacks have been taken. The authors develop an interesting approach that shows improvement on sequence diversity metrics relative to base line models.

**Weaknesses:**

The paper feels close to the acceptance threshold for me. However,I felt the authors could do a bit more. The increase in seq diversity generated by RL-DIF is modest in some cases. The authors could provide (A) more information on the trade off between structural similarity and sequence diversity along the training trajectory (b) More information on how hyper parameter can be used to relax structural similarity constant to increase diversity (c) exploration of regularization strategies like entropy based regularization to increase the diversity of sequences generated about the gains shown. (d) validation of structural similarity of generated sequences via alpha fold.

**Questions:**

Could the authors could provide (A) more information (plots) on the trade off between structural similarity and sequence diversity along the RL training trajectory (b) More information on how hyper parameter can be used to relax structural similarity constant to increase diversity (c) exploration of regularization strategies like entropy based regularization to increase the diversity of sequences generated about the gains shown. (d) validation of structural similarity of generated sequences via alpha fold.

---

> ### Author Response · Authors · 2024-11-15
> **Clarification question**
>
> Thank you for your review! We have a quick clarification question as we work on producing answers to your questions: when you refer to  "structural similarity", do you wish to see the similarity amongst generated structures (e.g. a form of structural diversity) or between the generated structures and the target structure (i.e. what we call "structural consistency" in the paper)?

---

> ### Author Response · Authors · 2024-11-22
> **Authors' response**
>
> > The paper feels close to the acceptance threshold for me. However,I felt the authors could do a bit more. The increase in seq diversity generated by RL-DIF is modest in some cases. The authors could provide (A) more information on the trade off between structural similarity and sequence diversity along the training trajectory (b) More information on how hyper parameter can be used to relax structural similarity constant to increase diversity (c) exploration of regularization strategies like entropy based regularization to increase the diversity of sequences generated about the gains shown. (d) validation of structural similarity of generated sequences via alpha fold.
>
> A:We have added Appendix A3 to study this, both along the training trajectory and across different models/sampling strategies. We compare foldable diversity (FD) with two other metrics: folding success (fraction of sequences that meet the sc-TM threshold) and folded diversity (FdD; sequence diversity among sequences meeting the threshold). Table S4 shows that folding success improves over the first 1k RL steps, while FD is maintained.
>
> B: Essentially the only training hyperparameters that can affect diversity are (1) sc-TM threshold and (2) number of RL steps. Profiling (1) is expensive (requiring multiple full retrainings), and lowering the threshold significantly would reward biologically inconsistent structures. In Table S4 we study the effect of (2), which does suggest that diversity increases initially, but then collapses with too many RL optimization steps.
>
> C: We did not explore adding an entropy regularization term in this work, but agree that could be a more direct way to boost diversity.
>
>
> D: We appreciate your suggestion to use AlphaFold for structural validation. While AlphaFold is generally recognized for its higher accuracy, we opted for ESMFold primarily due to its lower computational resource requirements. In our study, where we evaluated thousands of sequences, using AlphaFold would have been prohibitively expensive in terms of computational cost and time.
>
> We also have some confidence that evaluating with AlphaFold would not significantly change the deltas between benchmarked IF methods. TM-score between ESMFold and AlphaFold are correlated, especially if we are binarizing with a sc-TM threshold of 0.7 [1]. Therefore, the relative ranking of models in terms of foldable diversity would likely remain unchanged. Nonetheless, we agree with the reviewer that repeating this work with AlphaFold could be an important next step.
>
> [1] https://www.sciencedirect.com/science/article/pii/S0022283624001888#f0005 (Figure 1)
>
> Questions:
> > Could the authors could provide (A) more information (plots) on the trade off between structural similarity and sequence diversity along the RL training trajectory (b) More information on how hyper parameter can be used to relax structural similarity constant to increase diversity (c) exploration of regularization strategies like entropy based regularization to increase the diversity of sequences generated about the gains shown. (d) validation of structural similarity of generated sequences via alpha fold.
>
> We believe our response to the questions posed in Weaknesses addresses these questions.

---

> > ### Author Response · Authors · 2024-11-25
> > **Reviewer R2ES Followup**
> >
> > We would like to ask the reviewer if our response has resolved their outstanding comments. Given the deadline closing tommorow, we want to see if we could provide any more analysis or clarification to demonstrate the strength of our work.

---

> > > ### Author Response · Authors · 2024-12-01
> > > **Reviewer R2ES Followup**
> > >
> > > With the deadline approaching, we would like to ask the reviewer if our responses has resolved their outstanding comments? We can provide any more clarification if needed.

---

> > > > ### Author Response · Authors · 2024-12-02
> > > > **End of discussion period coming up tonight**
> > > >
> > > > With the deadline for discussion being tonight, we would like to ask the reviewer if our response has resolved their comments and if so, if they could adjust their score? If there is any more questions or doubts remaining, please ask and we can provide clarification.

---

### Official Review · Reviewer_L2jQ · 2024-11-04

**Soundness:** 3
**Presentation:** 3
**Contribution:** 2
**Rating:** 5
**Confidence:** 4

**Summary:**

The paper focuses on inverse folding, i.e., recovering a residue sequence that would fold into a given 3D backbone structure. The key motivation for the paper is specifically to improve diversity of generated sequences. The method consists of a discrete diffusion model that is then fine-tuned using RL (policy gradient) to improve foldability (sc-TM). Evaluation includes a number of baseline methods, including ProteinMPNN, PiFold, etc.

**Strengths:**

The paper is clearly written and the evaluations are fairly extensive, involve a number of relevant baselines, and seem conducted with care.

**Weaknesses:**

The proposed method is largely a direct combination of available techniques. Discrete diffusion component seems to be a variant of GradeIF/D3PM with small changes, and the RL fine-tuning is based on the rather straightforward policy gradient method (e.g.) from Black et al. I don't really see generalizable methodological innovations. Of course, it's fine to mix and match methods with adjustments to produce a well-performing technique but then the paper rests on its empirical results.

I think it would be better to replace eq (8) with a measure calculated only within foldable sequences so as to separate foldability from diversity (so they can be assessed separately and together). Diversity was taken as the key motivation for the paper but the two notions (diversity and foldability) are now mixed together in the proposed metric. E.g., the downward slope in Figure 2 is in part just due to the fact that higher folding threshold automatically decreases the value of the current metric. Figure 2 can also give a wrong impression that RL-diff increases diversity since the metric can be improved by increasing foldability instead. In fact, the reward used for fine-tuning the diffusion model is sc-TM so tailored to increase foldability (not control diversity).

**Questions:**

The expectation after eq (3) should really be a sum

q(S_t|S_{t-1},v) should be q(S_{t-1}|S_t,v) in eq (9)

In the model architecture, line 183, do you mean p(s_0|s_t) rather than p(s_{t+1}|s_t) as currently written?

Table 1 would be easier to read if summarized in terms of curves. For example, calculate diversity within all generated samples from a method that pass a sc-TM threshold and then vary this threshold.

---

> ### Author Response · Authors · 2024-11-15
> **Clarification question**
>
> Thank you for your review! We are working on thorough responses to all your questions, but would like to quickly clarify the question re: Table 1 - we believe Figure 2 shows what you are requesting (curves of diversity as scTM threshold is varied, across all benchmarked methods). Could you clarify if there is something else you'd like to see?

---

> ### Author Response · Authors · 2024-11-22
> **Authors' response**
>
> > The proposed method is largely a direct combination of available techniques. ... Of course, it's fine to mix and match methods with adjustments to produce a well-performing technique but then the paper rests on its empirical results.
>
> We thank the reviewer for their comment and would like to take this opportunity to clarify our contributions:
>
> (1) While this work does combine a number of prior methods, we found structural consistency to be a difficult reward to optimize. In general, the methods we were building on top of did not work out of the box. To summarize the problem-specific optimizations:
>   * Removing SASA features significantly hampered the GradeIF architecture (see Appendix A1). We came up with a variant of the PiFold that supported discrete diffusion to mitigate this regression.
>   * Our experiments showed that the D3PM hybrid loss out-performed GradeIF’s cross-entropy loss
>   * We found the optimization to be extremely sensitive to DDPO hyperparameters and reward standardization, requiring careful tuning
>
> It is a generally open problem of how to design diverse protein sequences meeting complex quality criteria. In this paper, we aim to demonstrate that a careful application of these prior methods results in a model optimized for one of the most fundamental of those criteria (structural consistency)
>
> (2) We believe an important contribution of our work is the introduction of foldable diversity. IF methods have been primarily benchmarked using sequence recovery, sequence diversity, and structural consistency. Our work shows that these metrics alone do not fully capture the main purpose of an inverse folding model: sampling diverse sequences that fold into a target structure. We demonstrate an inherent tradeoff between the foldability of model-generated sequences and the diversity of those sequences. Foldable diversity is a metric designed to measure this; models that are less sensitive to the tradeoff have a better foldable diversity.
>
> (3) We explore scaling properties of inverse folding models in an open-source way. In particular, we report that a careful scaling of pretraining data, model size, and on-policy training enables RL-DIF to achieve competitive performance with much larger models trained on far more data. To enable future work and direct comparison by the research community, we feel it is important to release the IDs of the proteins used in training our models.
>
> > I think it would be better to replace eq (8) with a measure calculated only within foldable sequences so as to separate foldability from diversity (so they can be assessed separately and together). Diversity was taken as the key motivation for the paper but the two notions (diversity and foldability) are now mixed together in the proposed metric. E.g., the downward slope in Figure 2 is in part just due to the fact that higher folding threshold automatically decreases the value of the current metric. Figure 2 can also give a wrong impression that RL-diff increases diversity since the metric can be improved by increasing foldability instead. In fact, the reward used for fine-tuning the diffusion model is sc-TM so tailored to increase foldability (not control diversity).
>
> We thank the reviewer for pointing out this important point. This is indeed something we considered as we constructed the foldable diversity (FD) metric, and have added Appendix A3 to study your proposed metric (which we call folded diversity (FdD)) in detail.
> Our observations demonstrate the expected trade-off (FdD can be increased at the expense of folding success). But we consider FD to be a more grounded metric because it measures diversity in the realistic scenario of a fixed sampling budget. FdD measures the diversity among good sequences, after rejecting bad samples. This can be maximized by sampling from a uniform distribution over sequences and simply rejecting those that fail to fold (some settings in Table S2 have a 90+% fail rate).
>
>
> > since the metric can be improved by increasing foldability instead
> In general, we observe IF models increase foldability at the expense of diversity, which does not improve FD alone. RL-DIF is less sensitive to this tradeoff and consistently increases the folding success over DIF-Only from anywhere from 2% to 19% while maintaining a competitive foldable diversity.
>
> Questions:
> > The expectation after eq (3) should really be a sum
>
> > q(S_t|S_{t-1},v) should be q(S_{t-1}|S_t,v) in eq (9)
>
> > In the model architecture, line 183, do you mean p(s_0|s_t) rather than p(s_{t+1}|s_t) as currently written?
>
> Thank you for catching these typos - fixed in the manuscript.
>
> > Table 1 would be easier to read if summarized in terms of curves. For example, calculate diversity within all generated samples from a method that pass a sc-TM threshold and then vary this threshold.
>
> We believe Figure 2 shows the requested curves, but are happy to iterate on the figure if we have misunderstood this point.

---

> > ### Comment · Reviewer_L2jQ · 2024-11-24
> >
> > I do not find that "foldable diversity" is a reasonable metric to adopt for comparisons as it confounds two key metrics that could be easily demonstrated jointly. I certainly agree that folded diversity alone does not suffice. However, it seems more insightful to show a 2D plot of folded diversity vs folding success, two easily understandable metrics, calculated at different thresholds to result in a few points on this plot per method. This would give small curvelets moving to the left as increasing the threshold decreases folding success. Comparing methods in this manner would directly highlight how the two dimensions are balanced (analogously to precision recall). Much of this is calculated in a table form in the appendix and should really replace figure 2 in the paper. It would also more clearly highlight the effect of RL fine-tuning that is based on increasing folding success only.
> >
> > My main concern of limited methodological advances in the paper remains. I will maintain the score for this reason.

---

> > > ### Author Response · Authors · 2024-11-25
> > > **Response to Reviewer L2jQ**
> > >
> > > We thank the reviewer for their comment. We agree that these metrics are more straightforward to understand and better demonstrate the effect of the RL fine-tuning in the increase in folding success. We will augment figure 2 with the plots described by the reviewer showing this curve as we change the threshold to demonstrate, as is already shown in the supplemental tables, the strength of our approach relative to other state-of-the-art models.
> > >
> > > Regarding methodological innovations, we believe our empirical results underscore the value of our work:
> > >
> > > (1) RDIF/DIF-Only achieves state-of-the-art results across all evaluated datasets.
> > >
> > > (2)Scaling RLDIF/DIF-Only delivers competitive performance with ESMIF, using 18× fewer parameters and 48× less data.
> > >
> > > Additionally, we release the pretraining data to enable broader use and further development of inverse folding models—an openness not matched by ESMIF, which did not release its training code and dataset. As the reviewer highlighted in their first review, our empirical results can compensate for any lack of methodological innovation.

---

> > > > ### Author Response · Authors · 2024-12-02
> > > > **End of discussion period coming up**
> > > >
> > > > We thank the reviewer for their continued engagement with our work. With the end of the discussion period coming up we wanted to ask and see if our comments resolved their concerns over methodological advances. As we stated earlier, we believe our empirical results underscore the value of our work to the community. Moreover, we wanted to see if there were any lingering questions or concerns that we could address before the discussion period ends.

---

### Meta-Review · Area_Chair_rpJX · 2024-12-20

**Metareview:**

This paper constructs a discrete diffusion model combined with RL fine tuning to solve the inverse protein folding problem, with a particular focus on high "foldable diversity": sequences that are dissimilar to each other but nevertheless fold into the correct target. The reviewers shared concerns about a lack of methodological novelty (L2jQ, 6Vm7), and concerns about the mixing of foldability and diversity (L2jQ) when these could be separated and analyzed as a trade-off, which 6Vm7 also points out in the author discussion period. Reviewer R2ES similarly asks for data on the trade off between structural similarity and sequence diversity. Ultimately, ignoring for the moment novelty concerns, which are often vague and hard to tackle concretely, I think the paper would be much improved by taking to hear the unanimous point about trade-offs in these metrics, and plotting these metrics as axes in a 2D plot against each other rather than reporting or even combining them.

**Additional Comments On Reviewer Discussion:**

I actually do think the authors raise good points and counterarguments to the weaknesses raised by the reviewers that I discuss above, but it ultimately seems like more time is needed to develop these into "show don't tell" arguments. In particular:

- On the topic of novelty: if the methods you are building on top of don't work out of the box, consider showing this in the paper, e.g. by comparing to these "out of the box" methods via an ablation study, where you show that, as you remove components of your own method and revert to these existing techniques significant performance is lost.

- On the topic of metric trade-offs: I think it's good to have promised the paper be updated with some of these results, but it's challenging to make a judgement call on accepting the paper now without them available.

---

### Decision · Program_Chairs · 2025-01-22

Reject